HSP90B1 regulates autophagy via PI3K/AKT/mTOR signaling, mediating HNSC biological behaviors

Li Chao
Lin Xiaoyu
Su Jiping ymsu2@126.com
Department of Otolaryngology-Head and Neck Surgery, First Affiliated Hospital of Guangxi Medical University , Nanning , China
Sistla Srinivas
Electronic publication date: 2024 Apr 5
Publication date: 2024
Volume: 12
Electronic Location ID: e17028
Received 2023 Nov 17; Accepted 2024 Feb 7
Copyright: ©2024 Li et al.
Copyright year: 2024
Copyright holder: Li et al.
License: This is an open access article distributed under the terms of the Creative Commons Attribution License, which permits unrestricted use, distribution, reproduction and adaptation in any medium and for any purpose provided that it is properly attributed. For attribution, the original author(s), title, publication source (PeerJ) and either DOI or URL of the article must be cited.
License URL: https://creativecommons.org/licenses/by/4.0/

Keywords: HSP90B1, HNSC, Autophagy, PI3K/AKT/mTOR signaling, Biological behavior

Funding: The authors received no funding for this work.

==============================
Background

Autophagy, a crucial cellular mechanism, facilitates the degradation and removal of misfolded proteins and impaired organelles. Recent research has increasingly highlighted the intimate connection between autophagy and heat shock proteins (HSPs) in the context of tumor development. However, the specific role and underlying mechanisms of heat shock protein 90 beta family member 1 (HSP90B1) in modulating autophagy within head and neck squamous cell carcinoma (HNSCC) remain elusive.

Methods

Quantitative real-time PCR (qRT-PCR), Western blot (WB), immunohistochemistry (IHC) were used to detect the expression in HNSC cell lines and tissues. The relationship between HSP90B1 and clinicopathologic features was explored based on TCGA (The Cancer Genome Atlas) data and IHC results. The biological functions of HSP90B1 were analyzed through in vitro and in vivo models to evaluate proliferation, migration, invasion, and autophagy. The mechanisms of HSP90B1 were studied using bioinformatics and WB.

Results

HSP90B1 was upregulated in HNSC cells and tissues. High HSP90B1 levels were associated with T-stage, M-stage, clinical stage, and poor prognosis in HNSC patients. Functionally, HSP90B1 promotes HNSC cell proliferation, migration, invasion and inhibits apoptosis. We discovered that HSP90B1 obstructs autophagy and advances HNSC progression through the PI3K/Akt/mTOR pathway.

Conclusion

Our study demonstrates that HSP90B1 is highly expressed in HNSC. Furthermore, HSP90B1 may regulate autophagy through the PI3K/Akt/mTOR pathway, mediating HNSC cell biological behaviors. These provide new insights into potential biomarkers and targets for HNSC therapy.

Introduction

Head and neck squamous cell carcinoma (HNSC) is the sixth most common cancer worldwide (Fang et al., 2018). It is responsible for nearly 500,000 new cases annually, with a mortality rate of approximately 20% among these patients (Torre et al., 2015; Siegel, Miller & Jemal, 2015). Despite progress in combination therapy, the five-year survival rate for HNSC patients remains static at around 50% (Nohata et al., 2011). This highlights the critical need for deeper insights into the molecular mechanisms of HNSC carcinogenesis to improve early diagnosis, prognostic predictions, and the formulation of more efficacious treatment strategies.

HSP90 is the most abundant class in HSPs, whose expression could reach up to 4%–6% under stress conditions, and about 600 clients have been found in mammal (Prodromou, 2016; Patel et al., 2011). Central to this research is Heat Shock Protein 90 β Family Member 1 (HSP90B1), also known as Grp94. This molecular chaperone is intricately involved with numerous signaling proteins, often found mutated, chimeric, or overexpressed in various cancers (Eletto, Dersh & Argon, 2010). HSP90B1 interacts with over 100 different client proteins (Birbo et al., 2021; Whitesell & Lindquist, 2005), underscoring its substantial role in cancer biology. Beyond its response to heat stress, HSP90B1 plays a significant part in adapting to diverse tumor microenvironmental factors such as glucose deficiency (Pugh et al., 2022), hypoxia (Byun et al., 2023), acidosis (Boussadia et al., 2018), and immune stimulation (Lin et al., 2020). Its impact on the progression of various cancers like lung adenocarcinoma (Duan & Xin, 2020), bladder cancer (Fang et al., 2019), breast cancer (Sundaram, Kc & Uludağ, 2022), and myeloma (Usmani et al., 2010), suggests its potential as a therapeutic target. Further investigation is necessary to understand HSP90B1’s mechanism of action in tumors.

Recently, autophagy has gained recognition for its essential role in tumor progression. Intriguingly, HSP90B1 has been identified to play a special role in this process at the same time (Abbonante et al., 2023; Santos et al., 2023). Autophagy is critical for cellular survival and metabolic equilibrium, facilitating protein degradation and organelle turnover to preserve intracellular and tumor microenvironmental homeostasis (Santos et al., 2023). It is a catabolic process that intricately regulates cancer behavior and treatment sensitivity (Miller & Thorburn, 2021). Autophagy is regulated in a variety of ways, especially PI3K/AKT/mTOR pathway. The PI3K/AKT/mTOR/autophagy pathways play a crucial role in regulating tumor progression and maintaining the balance of the tumor microenvironment (Sun et al., 2021; Wu et al., 2020; Yoshida, 2017; Zhang et al., 2011). Nevertheless, the research into the regulatory interplay between HSP90B1 and autophagy remains sparse, with the specific mechanisms of HSP90B1 action in HNSC yet to be fully elucidated.

This research endeavors to bridge the knowledge gap by examining HSP90B1 expression in HNSC and delving into its functional mechanisms through both in vitro and in vivo studies. The hypothesis posits that HSP90B1 mediates autophagy via the activation of the PI3K/AKT/mTOR signaling pathway, thereby impacting HNSC cell proliferation, migration, invasion, and the inhibition of apoptosis. A deeper comprehension of this interaction could unveil novel aspects of HNSC molecular dynamics and pave the way for specialized therapeutic approaches.

Materials and Methods

Data source

Differential gene expression analysis in head and neck squamous cell carcinoma was conducted using the GEPIA2.0 database (Gene Expression Profiling Interactive Analysis 2.0, http://gepia2.cancer-pku.cn/#index), which is built upon the TCGA database (The Cancer Genome Atlas, https://portal.gdc.cancer.gov/). The selected tissue samples comprised 519 tumor and 44 paraneoplastic specimens. The inclusion criteria for the samples necessitated comprehensive patient history documentation, clinicopathological evaluation, availability of sufficient tissue for analysis, and consent for genomic study participation. The marked disparity between the quantity of tumor and normal samples could be attributed to challenges in sample collection, the focus of research, and the accessibility of data, among other factors. For prognosis analysis based on molecular expression, we employed the Kaplan–Meier Plotter (https://kmplot.com/analysis/). Additionally, clinical data and gene expressions of HNSC patients were retrieved from TCGA to discern gene-clinical correlations. The potential biological functions of HSP90B1 were investigated by downloading coexpression gene data from GEPIA2.0 and cross-referencing it with positively or negatively correlated gene sets from the UALCAN database (http://ualcan.path.uab.edu/index.html). KEGG pathway (https://www.genome.jp/kegg/kegg1a.html) and Wikipathway (https://www.wikipathways.org/) was performed for function enrichment analysis using Sangerbox (http://Sangerbox.com). These databases allowed researchers to download and analyze public datasets for scientific purposes.

Tumor samples

Tissue specimens were obtained from 45 HNSC patients treated in the Department of Otorhinolaryngology Head and Neck Surgery at the First Hospital of Guangxi Medical University between June 2022 and August 2023. Among these patients, 10 were diagnosed with oral squamous cell carcinoma (OSCC), 28 with laryngeal squamous cell cancer (LSCC), and seven with hypopharyngeal squamous cell cancer (HSCC). A total of 45 pairs of HNSC tissues and adjacent non-tumor tissues were collected with informed consent and the approval of the Institutional Ethics Board of the First Affiliated Hospital of Guangxi Medical University (NO.2022-KY-E-(195)). Inclusion criteria were stringent: patients must have had complete medical records, not have received chemotherapy or radiotherapy prior to surgery, and received a perioperative diagnosis of squamous cell carcinoma. Ethical compliance was ensured as informed consent was obtained from all participants, who signed the informed consent form approved by the Ethics Committee of the First Affiliated Hospital of Guangxi Medical University.

Cell cultivation

Normal human oral keratinocytes (HOK), laryngeal squamous cell cancer cells (TU686), hypopharyngeal squamous cell cancer cells (FaDu), and tongue squamous cell carcinoma cells (SAS) were acquired from Shanghai WHELAB Bioscience Co., Ltd (Shanghai, China). TU686 cells were maintained in RPMI 1640 medium (Gibco, Invitrogen, Carlsbad, CA, USA), FaDu cells in MEM medium (Gibco, Invitrogen, Carlsbad, CA, USA), and SAS cells in DMEM medium (Gibco, Invitrogen, Carlsbad, CA, USA). All media were supplemented with 10% fetal bovine serum and 1% penicillin-streptomycin. The cell cultures were incubated at 37 °C in a 5% CO2 atmosphere.

RNA extraction and quantitative real-time fluorescence quantitative PCR

Total RNA was isolated from cells and specimen tissues using Trizol reagent (Beyotime Scientific, China). RNA concentration was measured with a NanoDrop2000 (Thermo Fish Scientific, USA). RNA is stored at −80 °C. The expression level of HSP90B1 was determined by quantitative real-time PCR (qRT-PCR) (QuantStudio 6 Flex, Thermo Fisher Scientific, Waltham, MA, USA) using a reverse transcription kit (TransGen Biotech, Peking, China). Primers used were as follows: HSP90B1, 5′-CGAAGTTGGACAGTGGTAAAGAG-3′(forward), 5′-TGCCCAATCATGGAGATGTCT-3′(reverse); GAPDH, 5′-AATCCCATCACCATCTTCCAG-3′(forward), 5′-GAGCCCCAGCCTTCTCCAT-3′(reverse).

Protein western blot analysis

Cells were harvested for protein extraction, and the protein concentration was quantified using the bicinchoninic acid (BCA) method (Biosharp Biotech, Shanghai, China). Proteins were separated by electrophoresis, transferred onto polyvinylidene difluoride (PVDF) membranes (GE Healthcare, Chicago, IL, USA), and then the membranes were blocked with 5% skimmed milk for 1.5 h at room temperature. Primary antibodies were applied and incubated overnight at 4 °C. Subsequently, membranes were incubated with the appropriate secondary antibodies for 1.5 h at room temperature, followed by three washes with TBST buffer. Protein detection was performed using an Odyssey infrared imaging system, and signal intensity was quantified using ImageJ software. The primary antibodies used included: HSP90B1 (1:1000, rabbit, Biolab Biotech, China), CASP3 (1:1000, rabbit, KleanAB Scientific, China), BCL2 (1:1000, rabbit, Zenbio Biotech, China), Bax (1:1000, rabbit, Zenbio Biotech, China), P62 (1:1000, rabbit, Proteintech Group, China), LC3B (1:1000, rabbit, Zenbio Biotech, China), and GAPDH (1:2000, mouse, Zenbio Biotech, China). Secondary antibodies included anti-rabbit (1:15000, Zenbio Biotech, China) and anti-mouse (1:15000, Zenbio Biotech, China).

Immunohistochemistry

A total of 45 pairs of tissue samples were prepared for immunohistochemistry (IHC). The tissues were sectioned into 3 µm slices, dewaxed, rehydrated, and treated with 3% hydrogen peroxide for one hour to block endogenous peroxidase activity. Following antigen retrieval, the sections were incubated with an HSP90B1 primary antibody (1:1000, rabbit, Biolab, China) overnight at 4 °C. This was succeeded by an incubation with a secondary antibody (ZB-2305, ZSGB-BIO, CN) for one hour at room temperature. For visualization, 3,3’-diaminobenzidine (DAB) reagent (ZLI-9018, ZSGB-BIO, Beijing, China) was applied to develop the peroxidase reaction, followed by hematoxylin counterstaining. Microscopic images were captured using an Olympus C-5050 camera. Independent and blind evaluations of the staining results were performed by two pathologists who categorized the immunohistochemical outcomes based on the intensity of staining in positive cells and the proportion of positive staining. The expression level of HSP90B1 was thus determined.

Transient transfection and establishment of stable cell lines

For the transient transfection of HNSC cell lines, si-HSP90B1 RNAs and control RNAs were synthesized (WZ Biosciences, Shandong, China) and introduced into cells using the Lipofectamine 2000 reagent (Thermo Fisher, Waltham, MA, USA) as per the provided guidelines. The shHSP90B1 lentiviral vectors were created by choosing an appropriate siRNA (small interfering RNA) sequence. Following puromycin selection, cells with stable transfection were employed in subsequent experiments. In a parallel approach, HSP90B1 overexpression was accomplished in cell lines using an HSP90B1 overexpression plasmid (HSP90B1-OE) and a corresponding empty vector control (HSP90B1-EV). Sequence of siHSP90B1: 5′-TCGCCTCAGTTTGAACATTGA-3′(palindromic sequence), 5′-AAGTTGATGTGGATGGTACAT-3′(antisense sequence). Overexpression sequence:pLV[Exp]-CMV>LPCAT1[NM_024830.5 ] -CBH>GFP-2A-Puro (Cas9X™, Haixing Biosciences, China).

Cell Counting Kit-8 assays

Post-transfection, cells were centrifuged and counted. In a 96-well plate, each well received 1,000 cells in 100 µL of medium. The experiment was divided into four groups: the siHSP90B1 group(si) and its control group (siNC), overexpression group (OE) and empty plasmid vector group (EV), wild group,with each group having 5 replicate wells per day. Daily, 10 µL of CCK-8 reagent (Invitrogen™, Thermo Fisher, Waltham, MA, USA) was added to each well at a consistent time and incubated for 2 h in the same environment. The optical density (OD) of the cells was then measured at 450 nm.

Clonogenic assay

In these colony formation assays, cells were plated in six-well plates at a density of 1,000 cells per well and cultured for two weeks. Following the cultivation period, the medium was discarded, and the established colonies were fixed with a 4% paraformaldehyde solution. The fixed colonies were then stained with a 0.1% crystal violet solution. Photographs of the stained six-well plates were taken to document the colony formation, and the number of colonies was quantitatively assessed.

Transwell assay

Dilute the matrix gel with serum-free medium at a 1:29 ratio. Take 100 µL of the diluted gel, spread it on the upper chamber of the Transwell (Corning, NY, USA), and incubate for 1.5 h. Following transfection, 5  × 104 cells were placed in the upper chamber, and 600 µL of complete medium (containing 10% serum DMEM) was added to the lower chamber. The cells were cultured for 24 h, fixed with 4% paraformaldehyde, stained with 1% crystal violet, photographed under the microscope, and counted.

Wound healing experiment

Following transfection, the cells were pelleted by centrifugation and subsequently enumerated. Scratch inserts by ibidi GmbH, Germany, were affixed to twelve-well plates, and three replicates per group were established for si, siNC, wild, EV, and OE treatments. A consistent volume seeding 30,000 cells was placed into the insert’s designated compartment. Post-attainment of full confluence within the chambers, and robust adherence to the walls, the scratch inserts were carefully extracted. The resultant cellular monolayers were then transferred to six-well plates containing a serum-free growth medium for further culture. At the outset and after twenty-four hours, the wounds induced by the insert removal were examined using microscopy, and corresponding images were captured. The rate of wound closure, indicative of the healing rate, was quantified at the conclusion of the observation period.

Flow cytometric assessment

The supernatant was saved and the cells were treated with EDTA-free trypsin. The trypsin was neutralized with the saved supernatant and the cell suspension was collected. It was then washed twice with PBS and centrifuged at 1,000 rpm for 5 min. Cells were suspended in buffer containing Annexin V-Alexa Flour647/7-AAD Kit (4A Biotech, Beijing, China), and stained for apoptosis detection using flow cytometry.

In vivo experiments

For constructing a model of hematogenous metastases, TU686 cells (1 × 107) were subcutaneously injected into the caudal vein of the mice. The animal experiments conducted in this study were approved by the Ethics Committee of Guangxi Medical University (approval number: 202302003). We obtained male BALB/c nude mice of SPF grade from the school’s animal facility, which were 5 weeks old, to create subcutaneous tumorigenic and hematogenous metastatic tumor models. TU686 cells were transfected with lentivirus containing shHSP90B1 or shNC. For constructing a model of Subcutaneous tumor, TU686 cells (5  × 106) were subcutaneously injected into the armpits of the mice.For constructing a model of hematogenous metastases, TU686 cells (1 × 107) were subcutaneously injected into the caudal vein of the mice. These procedures are performed without anesthesia because of the small injection volume of cell fluid (100–150 ul), short injection duration, and skilled injection technique. All mice were kept in animal facilities without pathogens. The animal room was kept at 18–23 °C, the humidity was 40%–60%, and the light and shade cycle was 10–14 h. In order to get the most scientific experimental results and according to the principle of 3R reduction, there are six mice in each group, a total of 12 mice were randomly divided into two groups. All food, cages, water and other items that came into contact with rats were sterile, and they were treated in certified biosafety cabinets using aseptic technology. Tumor size were measured every 7 days. On day 28, the maximum tumor diameter was about 15 mm. Thus, all mices of ubcutaneous tumorigenic groups were euthanized. Then, tumor size and weight were measured,and tumor tissues were harvested for immunohistochemistry. On day 48 (when fluorescence was detected in all mice of shNC groups), detecting visceral hematogenous metastases by animal in vivo imaging. During the experiment, the mices showed no cachexia and abnormal pain, all mices were killed by cervical dislocation after anesthesia, while those with tumors injected into the tail vein were observed for 4 months. Cervical dislocation euthanasia after anethetized; after anesthetizing the mice with isoflurane, the experimenters grabbed the base of the tail and lifted it up, placed it on the cage cover or other rough surface, and pressed down hard on the head and neck with the thumb and index finger of the other hand, the right hand grabbed the base of the tail and pulled it backward and upwards, causing the dislocation of the cervical spine and the separation of the spinal cord from the brain stem. The experimental animals died immediately.

Statistical analysis

The same type of experiment was repeated three times. Each data that needs manual measurement will be repeated three times, and the average value will be taken as the result. Data were analyzed using SPSS 23.0 software, and measurements were presented as mean ± standard deviation (X + SD). The correlation of clinicopathological characteristics was analyzed by chi-square test and Log-rank statistical analysis, and all experimental and control groups were statistically analyzed by independent samples T-test, and the difference was considered statistically significant at p < 0.05.0.01 <*p < 0.05, 0.001 < **p < 0.01, ***p < 0.001.

Results

HSP90B1 is overexpressed in HNSC and correlated with poor prognosis in HNSC patients

To examine the differential expression of HSP90B1, we utilized the GEPIA database (http://gepia.cancer-pku.cn/) and performed analyses on both cell lines and tissue specimens. Data from GEPIA indicated that HSP90B1 expression was elevated in cancerous tissues relative to their normal counterparts (Fig. 1A). Furthermore, PCR analysis of cell lines revealed pronounced upregulation of HSP90B1 in head and neck squamous cell carcinoma (HNSC) cells (TU686 and Fadu, SAS) when contrasted with normal human oral keratinocytes (HOK) (Fig. 1B). Similarly, HSP90B1 displayed increased expression in HNSC tissue samples compared to adjacent non-tumorous tissues from our patient cohort (Fig. 1C). Western blot assays corroborated the heightened expression of HSP90B1 in HNSC cell lines (Fig. 1D). Immunohistochemistry (IHC) assays further verified the upregulation of HSP90B1 in cancer tissues versus normal tissues (Fig. 1E). Prognostic evaluation using the Kaplan–Meier plotter database revealed an inverse relationship between elevated HSP90B1 expression in HNSC patients and their overall survival (OS), suggesting an adverse prognostic impact (Fig. 1F). Collectively, these findings denote that HSP90B1 is overexpressed in HNSC and may serve as a biomarker for unfavorable prognosis.

Figure 1 High expression of HSP90B1 in head and neck squamous carcinoma is associated with poor prognosis.

(A) HSP90B1 expression levels in mRNA. GEPIA database. (B) PCR assays were used for cell lines (normal human oral keratinocytes (HOK), laryngeal squamous cell cancer cells (TU686), hypopharyngeal squamous cell cancer cells (FaDu), and tongue squamous cell carcinoma cells (SAS)). (C) HNSC tissue assays (oral squamous cell carcinoma (OSCC), laryngeal squamous cell cancer (LSCC), hypopharyngeal squamous cell cancer (HSCC)). (D) WB was used to determine the expression level of HSP90B1 in cell lines. (E) Immunohistochemical analysis was used to determine the expression level of HSP90B1 in HNSC tumors and paracancerous normal tissues, the large image is a magnification of 200×, 1000×. (F) HSP90B1 was analyzed using the Kaplan-Mier database for survival-related statistics in HNSC patients.

To delve deeper into the clinical relevance of HSP90B1 in HNSC, we sourced expression profiles of HSP90B1 and clinical data from the TCGA database for comprehensive analysis. Our analysis revealed that patients exhibiting elevated HSP90B1 levels experienced significantly reduced survival times. Notably, heightened HSP90B1 expression correlated with advanced clinical T Stage, the presence of distant metastasis, and higher tumor grade (Table 1). Further, by correlating immunohistochemistry (IHC) results with clinicopathological features, a significant association between HSP90B1 expression and clinical stage was established (Table 2).

Table 1 Association between LPCAT1 expression (based on mRNA expression) and clinicopathological factors.

Data source TCGA database.

	No.(%)	mRNA expression of HSP90B1	
Variable		M	SD (±)	P-value	
Survival_status					
Alive	283 (56.6%)	7.431	0.533	0.031	
Dead	217 (43.4%)	7.535	0.528		
Sex					
Male	181 (36.2%)	7.481	0.541	0.356	
Female	319 (63.8%)	7.531	0.503		
Stage					
I–II	110 (22%)	7.414	0.496	0.075	
III–IV	390 (78%)	7.517	0.539		
T stage					
T1–T2	177 (35.4%)	7.423	0.487	0.037	
T3–T4	323 (64.6%)	7.525	0.535		
N stage					
N0–N1	322 (64.4%)	7.463	0.552	0.083	
N2–N3	178 (35.6%)	7.548	0.485		
M stage					
M0	475 (95%)	7.476	0.528	0.001	
M1	25 (5%)	7.83	0.495		
Grade					
I–II	362 (72.4%)	7.454	0.535	0.004	
III–IV	138 (27.6%)	7.600	0.508		
Lymphovascular invasion					
Yes	281 (56.2%)	7.511	0.532	0.422	
No	219 (43.8%)	7.472	0.531		
Perineural invasion					
Yes	314 (62.8%)	7.472	0.522		
No	186 (37.2%)	7.529	0.547	0.247	

Table 2 Association between LPCAT1 expression (based on IHC) and clinicopathological factors.

Score the IHC results of human tissues and compare them based on clinicopathological characteristics.

Clinical characteristics			HSP90B1	Positive rate (%)	χ2	p	
Pathological classification		Total	Negative	Positive				
Sex	Male	40	9	31	77.5	0.094	0.759	
	Female	5	2	3	60.00			
Age	<60	18	5	13	72.22	0.180	0.671	
	≥60	27	6	21	77.78			
Tumor site	OSCC	9	1	7	77.78	0.506	0.477	
	HSCC	11	3	8	72.73			
	LSCC	25	7	19	76.00			
T stage	T1-T2	13	5	8	61.54	1.945	0.163	
	T3-T4	32	6	26	81.25			
N stage	N0-N1	20	7	13	65.00	1.265	0.261	
	N2-N3	25	4	21	84.00			
M stage	M0	45	11	34	75.56	-	-	
	M1	0	0	0	0			
Clinical stage	I-II	10	5	5	50	4.546	0.033	
	III-IV	35	6	29	80.00			
Differentiation
degree	Low- intermediate	28	5	20	71.43	1.224	0.268	
	High	17	6	11	64.71			

HSP90B1 enhance proliferation and inhibits apoptosis in HNSC cell

To elucidate the role of HSP90B1 in head and neck squamous cell carcinoma, we conducted knockdown and overexpression experiments on TU686 and SAS cell lines (Fig. 2C). The CCK8 assay indicated that cells with HSP90B1 knockdown (siHSP90B1 group, si) exhibited a lower proliferation rate compared to the siRNA negative control group (siNC), while cells with HSP90B1 overexpression (OE group) demonstrated higher proliferation than the empty vector control group (EV) in both TU686 and SAS cells (Figs. 2A, 2B). WB assays and flow cytometry were employed to investigate HSP90B1’s role in apoptosis inhibition in HNSC cells. The WB assays showed increased expression of pro-apoptotic markers CASP3 and BAX and decreased expression of anti-apoptotic BCL2 in the siHSP90B1 group. In contrast, the overexpression group exhibited the reverse pattern. Correspondingly, flow cytometry results revealed an increase in apoptotic cells in the siHSP90B1 group and a decrease in the overexpression group (Fig. 2D). These findings suggest that high expression of HSP90B1 promotes proliferation and inhibits apoptosis in HNSC cells.

Figure 2 HSP90B1 promotes HNS cell proliferation and inhibits apoptosis.

Initially, the knockdown and overexpression of HSP90B1 in cell lines were established. (A–B) The proliferative capacities of TU686 and SAS cells were evaluated using the CCK-8 and Clone assays. (C) HSP90B1 expression levels and apoptotic molecules were assessed using Western blot (WB). (D) Apoptosis level alterations were detected by flow cytometry, employing Fluor647/7-AAD double staining.

HSP90B1 promotes invasion, metastasis of HNSC cells

Cell migration and invasion are critical factors in the progression of HNSC. Wound healing assays demonstrated that HSP90B1 knockdown significantly increased the scratch width in the si group compared to the siNC at 12 h post-wounding. Conversely, cells with HSP90B1 overexpression exhibited decreasing trends (Fig. 3A). Furthermore, transwell assays indicated a decrease in the number of invasive cells in the si group compared to the siNC group, with an increase observed in OE group relative to the EV group (Fig. 3B). These findings indicate that elevated HSP90B1 expression enhances HNSC cell migration and invasion, contributing to the malignancy’s aggressive biological behavior.

Figure 3 HSP90B1 promotes HNSC invasion, metastasis.

Each type of cell was divided into five groups: si, siNC, Wild, EV, and OE. (A) Detection of TU686, SAS cell invasion ability using Transwell assay, image is enlarged 200. (B) Detection of cellular healing capacity using the wound healing assay.

HSP90B1 activates the AKT/mTOR pathway to inhibit autophagy

To explore the potential mechanisms of HSP90B1 in HNSC, we analyzed the top 200 genes closely related to HSP90B1 (similar genes) using the GEPIA database, identifying 1,613 positively correlated genes and 80 negatively correlated genes with a correlation coefficient greater than 0.3 from the UALCAN database. We intersected these datasets and pinpointed 75 genes consistently positively associated with HSP90B1. These genes underwent pathways enrichment analysis via Sangerbox (Fig. 4A). The analysis revealed significant enrichment of HSP90B1 such as focal adhesion, human papillomavirus infection, the PI3K/AKT signaling pathway, and ECM-receptor interactions (Fig. 4A). Notably, the PI3K/AKT signaling pathway emerged as the most prominent. Wikipath analysis also suggests that HSP90B1 participates in the PI3K/AKT/mTOR signaling pathway by regulating AKT phosphorylation (see Supplemental Information). Given that the PI3K/AKT/mTOR pathway is pivotal in regulating autophagy, we hypothesized that HSP90B1 influences autophagy through this pathway (Fig. 4B). Western blot analysis demonstrated that HSP90B1 knockdown reduced phosphorylated AKT (p-AKT), suggesting an inhibition of PI3K/AKT phosphorylation signaling (Fig. 4C). Meanwhile, P62 and LC3 are vital markers for autophagy process(Yu, Chen & Tooze, 2018), P62 levels decreased while LC3II/LC3I levels increased, indicating enhanced autophagy (Fig. 4C). In contrast, HSP90B1 overexpression resulted in inverse trends in these markers. These results indicate that HSP90B1 may suppress autophagy through the PI3K/AKT signaling pathway.

Figure 4 HSP90B1 activates the AKT-mTOR pathway to inhibit autophagy.

(A) Obtaining gene sets with similar functions to HSP90B1 from the GEPIA database, and gene sets with positive or negative correlation with HSP90B1 from UALCAN. Then, place common genes from the two databases into Sangerbox for pathway enrichment. (B) The WB assay was used to detect protein expression of p-mTOR, mTOR, p-AKT, and AKT in the cell lines. The corresponding statistics of protein expression are presented in the bar graph. (C) The WB technique was employed to ascertain the protein expression levels of P62 and LC3B in the cell lines.

HSP90B1 promotes HNSC cell proliferation and metastasis in vivo

To further understand the in vivo functional role of HSP90B1 in head and neck squamous cell carcinoma (HNSC), we generated a cell line with stable HSP90B1 knockdown (shHSP90B1) and a control cell line (shNC). We established both a subcutaneous tumorigenic model (Fig. 5A) and a tail vein hematogenous metastasis model in BALB/C nude mice (Fig. 5C). The study revealed that tumors from HSP90B1-depleted cells were notably smaller and exhibited slower growth compared to the control (Fig. 5A). Immunohistochemical analysis confirmed reduced HSP90B1 expression in the sh-HSP90B1 group (Fig. 5B). In the hematogenous metastasis model, the sh-HSP90B1 group showed significantly lower compositional density (Fig. 5C), indicating that HSP90B1 plays a role in promoting proliferation and metastasis of HNSC in vivo.

Figure 5 HSP90B1 promotes HNSC proliferation and metastasis in vivo.

TU686 cells, stably transfected with shNC and sh-HSP90B1, were administered to BALB/c nude mice through both subcutaneous and tail vein injections. After 28 or 48 days post-injection, the mice were euthanized via cervical dislocation under anesthesia. (A) Images at the top display the nude mice and their subcutaneous tumors, while data on tumor volume changes and final tumor weights are provided at the bottom. (B) Immunohistochemical analysis was conducted on subcutaneous tumor tissue sections from these mice. (C) Hematopoietic metastases were visualized using an animal live scan imager.

Discussion

The identification of pivotal genes is crucial for elucidating the pathways implicated in HNSC and for developing targeted therapies due to its aggressive nature. Our study reveals a significant overexpression of HSP90B1 in HNSC, correlating with unfavorable patient outcomes. Notably, HSP90B1 influences key cellular processes including proliferation, invasion, metastasis, and apoptosis in HNSC cells. Additionally, it impacts autophagy via the PI3K/AKT/mTOR signaling pathway.

HSP90B1 plays a pivotal role in regulating cancer cell survival and death by maintaining ER stress sensors, protein folding capacity, and inhibiting pro-apoptotic mechanisms (Kim, Cho & Lee, 2021). It acts as a chaperone for toll-like receptor (TLR) (Graustein et al., 2018), integrin subunit (Liu & Li, 2008), and wnt co-receptor low-density lipoprotein receptor-associated protein (LRP6) (Liu et al., 2013), which are implicated in tumor microenvironment and tumor cell stemness (Santos et al., 2023; Tang et al., 2021). HSP90B1 is also associated with the development of many diseases by mediating key proteins of PCD, including receptor interaction serine/threonine kinase (RIP) 1 in necroptosis (Hu et al., 2020), glutathione peroxidase (GPX) 4 in ferroptosis (Wu et al., 2019), and Beclin-1,mTOR in cell apoptosis and autophagy (Hasan et al., 2020). Consequently, targeting HSP90B1 and its co-chaperones may offer a promising therapeutic strategy against a broad spectrum of diseases (Zuehlke, Moses & Neckers, 2018). Our analysis of the GEPIA 2.0 database identified an upregulation of HSP90B1 expression in HNSC, Kaplan–Meier Plotter analysis revealed that HSP90B1 overexpression correlates with poorer patient outcomes. Further, tissue samples, cell line PCR, immunohistochemistry (IHC), and Western blot (WB) assays confirmed HSP90B1’s high expression. Clinical data from the TCGA database also indicated a significant correlation between HSP90B1 expression and T-stage, M-stage, and grading, aligning with findings by Chen, Feng & Chen (2022) in oral cancer. Additionally, our in vitro experiments demonstrated that HSP90B1 promotes HNSC cell proliferation, invasion, and metastasis, while suppressing apoptosis, confirming our hypothesis.

The occurrence of autophagy is a complex process and is essential in tumor progression (Miller & Thorburn, 2021). It sequesters damaged organelles, misfolded proteins, and mutated proteins within double-membrane vesicles known as autophagosomes. These autophagosomes then merge with lysosomes, facilitating the breakdown of their contained components. The conversion of microtubule-associated protein 1 light chain 3 (LC3I) to phosphatidylethanolamine-conjugated LC3-II, and LC3 II is integral to autophagosome membrane formation. In this process, p62 serves as an adaptor molecule, binding to ubiquitinated proteins and LC3-II on autophagosome membranes, resulting in their joint degradation. The ULK complex initiates the autophagic response, which can be suppressed by mTOR (Fig. 4B). The pathophysiologically-relevant PI3K/AKT/mTOR pathway, a classic pathway for tumor regulation, is also a critical target in tumor therapy due to its role in modulating autophagic activity (Itakura & Mizushima, 2010; Popova & Jücker, 2021; Xu et al., 2020). The unique role of autophagy in the continuous progression of tumor cells has attracted widespread attention from scholars. However, the precise impact of autophagy on cancer remains controversial, one perspective suggests that autophagy enables cellular adaptation to environmental stressors such as endoplasmic reticulum stress, hypoxia, oxidative stress, and supports the regulation of the cell cycle and apoptosis (Miller & Thorburn, 2021; Yu, Chen & Tooze, 2018). In contrast, other researchers argue that autophagy can restrain tumor development by enhancing intracellular degradation, maintaining cellular homeostasis, and preventing uncontrolled cell proliferation. Additionally, its potential to attenuate inflammation and minimize DNA damage has been linked to tumor suppression. Notably, the implications of autophagy are contingent on disease stage and genetic context (Debnath, Gammoh & Ryan, 2023). Qiu et al. (2023) research has found that enhanced autophagy is accompanied by limited proliferation of oral squamous cell carcinoma. Gao et al. (2020) demonstrated that disruptors of autophagy, specifically via the PI3K/AKT/mTOR pathway, contribute to tumor progression and chemoresistance in laryngeal squamous cell carcinoma.

Several studies have shown that HSP90B1 can documents the regulatory influence of HSP90B1 on the PI3K/AKT/mTOR signaling cascade (Li et al., 2012; Li et al., 2023), and may also suggesting its potential modulation of autophagy through core autophagic proteins ULK1 and BECN1 (Peng et al., 2022). The interaction between HSP90 and ULK1 contributes to the autophosphorylation of ULK1 at Ser1047, which is interrupted by the treatment of HSP90 inhibitor (Joo et al., 2011). HSP90 forms a complex with Beclin-1 through an evolutionarily conserved domain to maintaining its stability and phosphorylation (Gassen et al., 2014; Xu et al., 2011). Santos et al. (2023) identified HSP90B1 as a candidate endoplasmic reticulum stress-associated protein with a putative role in autophagy (Abbonante et al., 2023). Moreover, studies have hinted at a connection between HSP90B1 and autophagy (Santos et al., 2023; Xu et al., 2020), yet this relationship merits further exploration and corroboration.

Bioinformatics enrichment analysis of HSP90B1-associated genes suggests the involvement of HSP90B1 in the PI3K/AKT/mTOR signaling pathway in HNSC, corroborating the results obtained from Wikipathways (Supplemental Information 3). It is postulated that HSP90B1 also contributes to the modulation of autophagy. Subsequent Western Blot assays substantiated that HSP90B1 upregulates the PI3K/AKT/mTOR signaling pathway and concurrently suppresses autophagy.

Autophagy serves as a critical mechanism for ensuring cellular survival and functionality, and its distinctive role in cancer biology has garnered significant scholarly attention. Achieving precision in cancer therapy necessitates a more comprehensive understanding of autophagy. HSP90B1 influences tumor progression through various pathways, and elucidating its specific mechanisms in HNSC warrants additional investigation.

Conclusion

In summary, our study found that high levels of HSP90B1 expression in HNSC patients are associated with poor prognosis. HSP90B1 promotes the proliferation, invasion, and metastasis of HNSC while also inhibiting apoptosis. This suggests that HSP90B1 may regulate the biological behavior of HNSC by inhibiting autophagy through the PI3K/AKT/mTOR signaling pathway.

Supplemental Information

Supplemental Information 1 Raw data of WB results

Repeated 3 times.

Supplemental Information 2 Raw data

Supplemental Information 3 HSP90B1 in wikipathways

Wikipathway shows that HSP90B1 is involved in the PI3K/AKT/mTOR signaling pathway.

Supplemental Information 4 Author checklist

Additional Information and Declarations

Competing Interests

Author Contributions

Human Ethics

Animal Ethics

Data Availability

The authors declare there are no competing interests.

Chao Li conceived and designed the experiments, performed the experiments, analyzed the data, prepared figures and/or tables, and approved the final draft.

Xiaoyu Lin performed the experiments, analyzed the data, prepared figures and/or tables, and approved the final draft.

Jiping Su conceived and designed the experiments, authored or reviewed drafts of the article, and approved the final draft.

The following information was supplied relating to ethical approvals (i.e., approving body and any reference numbers):

Research Ethics Committee of First Affiliated Hospital of Guangxi Medical University (NO.2022-KY-E-(195)).

The following information was supplied relating to ethical approvals (i.e., approving body and any reference numbers):

Ethics Committee of Guangxi Medical University, approval number: 202302003.

The following information was supplied regarding data availability:

The raw data is available in the Supplemental Files.

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
