# Peer review of "HSP90B1 regulates autophagy via PI3K/AKT/mTOR signaling, mediating HNSC biological behaviors"

_PeerJ, doi:10.7717/peerj.17028_

## Round 0.1 · original submission · Major Revisions

I agree with the reviewers that the paper needs a major revision. To start with the second reviewer's comments on the "In Vitro Assays with HSP90B1 Overexpression" part I am convinced that this adds value to this paper. Additionally the changes suggested to the introduction enhanced the clarity of the manuscript.

A thoroughly simplified English writing is really appreciated. Figure legends must be self-explanatory. Gene Expression Profiling Interactive Analysis 2 analysis need a more thorough write-up as it brings in substantial differences between cancerous and normal data.

Please provide a rebuttal as and when you feel relevant and submit the revised version at the earliest.

**Language Note:** The Academic Editor has identified that the English language must be improved. PeerJ can provide language editing services - please contact us at copyediting@peerj.com for pricing (be sure to provide your manuscript number and title). Alternatively, you should make your own arrangements to improve the language quality and provide details in your response letter. – PeerJ Staff

Reviewer 1 ·

Basic reporting

This study found that expression of the heat shock protein HSP90B1 is increased in head and neck squamous cell carcinoma (HNSCC) compared to normal tissue. Higher HSP90B1 levels correlate with more advanced stage disease, metastasis, worse tumor grade, and poorer prognosis in HNSCC patients. Additionally, through in vitro and in vivo experiments, the authors demonstrate HSP90B1 promotes aggressive HNSCC behaviors including enhanced cell proliferation, invasion, and metastasis, while inhibiting cell death pathways like apoptosis. Furthermore, they present data suggesting a key mechanism is HSP90B1's activation of the PI3K/AKT/mTOR signaling cascade, which suppresses protective autophagy in cancer cells.

Overall the study design and findings appear novel and meaningful. The authors should address important and minor issues to strengthen the quality before the paper is considered for publication.

1. The manuscript requires substantial improvement in its writing. The authors need to comprehensively enhance the clarity and readability of the entire manuscript. Some notable issues as examples:
a) The abstract employs abbreviations such as RT-PCR, WB, IHC, and TCGA without prior clarification.
b) Numerous misspellings are evident, with the use of two different abbreviations, HNSC and HNSCC. In Line 96, ‘Weston’ is misspelled.
c) The introduction lacks sufficient detail. Additional information about HSP90B1 in cancer and the current state of research on autophagy should be incorporated.
d) Specifics regarding the type of PCR analysis conducted at Line 204 are not provided. Line128, 129, the sequence is same?
e) Concerning Lines 229-231, the assertion that high HSP90B1 expression promotes apoptosis appears incorrect.
f) In the result section, authors need to provide additional details to enhance clarity. For example, in Figure 4A, authors need to explain the significance of the varying numerical values. Additionally, Figure 1 A lacks explanations for the representations pertaining to OSCC, LSCC, and HSCC. Providing elucidations for these figures will contribute to a more comprehensive understanding of the results.

2. The structure and organization of the paper follows disciplinary norms. The sections are logically ordered.

3. Figures are relevant and informative. Raw data is provided as per journal requirements. However, the figure legend should have more additional details to enhance overall comprehension. Additionally, Figure 1A should be divided into separate figures, namely, Figure 1A, 1B, and 1C, to distinctly represent database, cell line, and tissue samples.

Experimental design

The data sourced from the GEPIA2.0 database requires additional information regarding the inclusion and exclusion criteria. Specifically, in Figure 1, the origin of the samples is unclear; while the number of T is indicated as 519, N is documented as 44. It is essential to clarify whether all samples are derived from the GEPIA database and elucidate the reason behind the substantial difference in numbers between T and N. Providing this clarification will enhance the transparency and understanding of the dataset.

Validity of the findings

no comments

Reviewer 2 ·

Basic reporting

1. Abstract Figure Addition: To visually underscore the molecular role of HSP90B1 within the PI3K/AKT/mTOR signaling pathway, the inclusion of an abstract figure would be highly beneficial. This would provide a clear, immediate understanding of the study's focal points.

2. Background and Introduction Enhancement:
- Autophagy and Cancer: There's a need for an expanded discussion on the background of autophagy and its implications in cancer within both the abstract and the introduction.
- Autophagy and Heat Shock Proteins: The relationship between autophagy and heat shock proteins should be clearly articulated.
- Focus on HSP90B1: A crucial aspect of the paper is the emphasis on HSP90B1 among heat shock proteins. It's imperative to elucidate the rationale behind selecting HSP90B1 for this in-depth analysis.

Experimental design

3. In Vitro Assays with HSP90B1 Overexpression: Given that most in vitro results currently rely on HSP90B1-knockdown assays, incorporating experiments with overexpressed HSP90B1 would strengthen the study. This approach could validate the detrimental effects of HSP90B1 in autophagy within cancer cells. While repeating all assays might not be necessary, selecting key experiments for this purpose is highly recommended.

Validity of the findings

4. Figure 1C and 5B: The scale bar is absent in these figures.

5. Labeling in Figures 4B and C: The labels for the negative control and knockdown groups in these figures seem to be missing.

---

## Round 0.2 · accepted · Accept

Reviewers have recommended this manuscript for publication.

Reviewer 1 ·

Basic reporting

no comment

Experimental design

no comment

Validity of the findings

no comment

Reviewer 2 ·

Basic reporting

The authors have addressed all my comments.

Experimental design

The manuscript is ready for publication.

Validity of the findings

See above.